# VTNet: Visual Transformer Network for Object Goal Navigation

**Heming Du**[1,3], **Xin Yu**[2]*\& **Liang Zheng**[1]
[1]Australian National University
[2]University of Technology Sydney
[3]CSIRO-DATA61
{heming.du, liang.zheng}@anu.edu.au, xin.yu@uts.edu.au

## Abstract

Object goal navigation aims to steer an agent towards a target object based on observations of the agent. It is of pivotal importance to design effective visual representations of the observed scene in determining navigation actions. In this paper, we introduce a Visual Transformer Network (VTNet) for learning informative visual representation in navigation. VTNet is a highly effective structure that embodies two key properties for visual representations: *First*, the relationships among all the object instances in a scene are exploited; *Second*, the spatial locations of objects and image regions are emphasized so that directional navigation signals can be learned. Furthermore, we also develop a pre-training scheme to associate the visual representations with navigation signals, and thus facilitate navigation policy learning. In a nutshell, VTNet embeds object and region features with their location cues as spatial-aware descriptors and then incorporates all the encoded descriptors through attention operations to achieve informative representation for navigation. Given such visual representations, agents are able to explore the correlations between visual observations and navigation actions. For example, an agent would prioritize "turning right" over "turning left" when the visual representation emphasizes on the right side of activation map. Experiments in the artificial environment AI2-Thor demonstrate that VTNet significantly outperforms state-of-the-art methods in unseen testing environments.

## 1 Introduction

The goal of target-driven visual navigation is to guide an agent to reach instances of a given target category based on its monocular observations of an environment. Thus, it is highly desirable to achieve an informative visual representation of the observation, which is correlated to directional navigation signals. In this paper, we propose a Visual Transformer Network (VTNet) to achieve an expressive visual representation. In our VTNet, we develop a Visual Transformer (VT) to extract image descriptors from visual observations and then decode visual representations of the observed scenes. Then, we present a pre-training scheme to associate visual representations with directional navigation signals, thus making the representations informative for navigation. After pre-training, our visual representations are fed to a navigation policy network and we train our entire network in an end-to-end manner. In particular, our VT exploits two newly designed spatial-aware descriptors as the key and query, (*i.e.*, a spatial-enhanced local descriptor and a positional global descriptor) and then encodes them to achieve an expressive visual representation.

Our spatial-enhanced local descriptor is developed to fully take advantage of all detected objects for the exploration of spatial and category relationships among instances. Unlike the prior work (Du et al., 2020) that only leverages one instance per class to mine the category relationship, our VT is able to exploit the relationship of all the detected instances. To this end, we employ an object detector DETR (Carion et al., 2020) since features extracted from DETR not only encode object appearance information, such as class labels and bounding boxes, but also contain the relations between instances and global contexts. Moreover, DETR features are scale-invariant (output from the

---
*Corresponding author

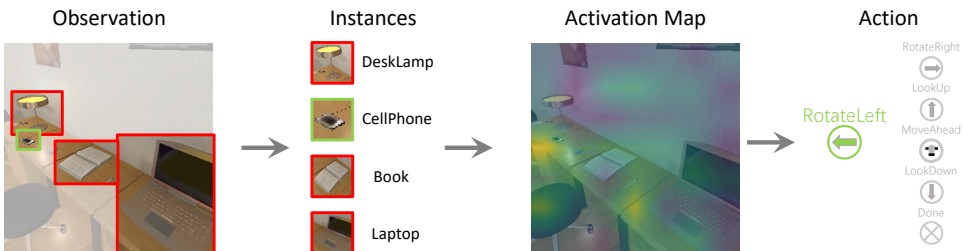

Figure 1: **Motivation of the Visual Transformer Network (VTNet).** A target class (*cellphone*) is highlighted by green bounding boxes. An agent first detects objects of interest from its observation. Then, the agent attends detected objects to the global observation by the visual transformer (VT). High attention scores are achieved on the *left side* of the observation, which correspond to the target (*cellphone*). Then, the agent will choose RotateLeft to reach targets.

same layer) in comparison to features used in ORG (Du et al., 2020). Considering that object positions cannot be explicitly decoded without the feed-forward layer of DETR, we therefore enhance all the detected instance features with their locations to obtain spatial-enhanced local descriptors. Then, we take all the spatial-enhanced local descriptors as the key of our VT encoder to model the relationships among detected instances, such as category concurrence and spatial correlations.[1]

Furthermore, we introduce a positional global descriptor as the query for our VT decoder. In particular, we associate the region features with image region positions (such as bottom and top) and thus facilitate exploring the correspondences between navigation actions and image regions. To do so, we divide a global observation into multiple regions based on spatial layouts and assign a positional embedding to each region feature as our spatial-enhanced global descriptor. After obtaining the global query descriptor, we attend the spatial-enhanced local descriptor to the positional global descriptor query to learn the relationship between instances and observation regions via our VT decoder.

However, we found directly training our VTNet with a navigation policy network fails to converge due to the training difficulty of the transformers (Vaswani et al., 2017). Therefore, we present a pre-training scheme to associate visual representations and directional navigation signals. We endow our VT with the capability of encoding directional navigation signals by imitating expert experience. After warming-up through human instructions, VT can learn instructional representations for navigation, as illustrated in Figure 1.

After pre-training our VT, we employ a standard Long Short Term Memory (LSTM) network to map the current visual representation and previous states to an agent action. We adopt A3C architecture (Mnih et al., 2016) to learn the navigation policy. Once our VTNet has been fully trained, our agent can exploit the correlations between observations and navigation actions to improve visual navigation efficiency. In the popular widely-used navigation environment AI2-Thor (Kolve et al., 2017), our method significantly outperforms the state-of-the-art. Our contributions are summarized as follows:

- We propose a novel Visual Transformer Network (VTNet) to extract informative feature representations for visual navigation. Our visual representations not only encode relationships among objects but also establish strong correlations with navigation signals.

- We introduce a positional global descriptor and a spatial-enhanced local descriptor as the query and key for our visual transformer (VT), and then the visual representations decoded by our VT are attended to navigation actions via our presented pre-training scheme, thus providing a good initialization for our VT.

- Experimental results demonstrate that our learned visual representation significantly improves the efficiency of the state-of-the-art visual navigation systems in unseen environments by 14.0% relatively on Success Weighted by Path Length (SPL).

---

[1]As observed in DETR, the number of objects of interest in a scene is usually less than 100. Thus, we set the key number to 100 in the VT encoder.

## 2 RELATED WORKS

Visual navigation, as a fundamental task in robotic and artificial intelligence, has attracted increasing attention recently. Traditional methods (Oriolo et al., 1995) often leverage environment maps for navigation and divide a navigation task into three steps: mapping, localization and path planning. Some approaches employ a given map to obviate obstructions (Borenstein & Koren, 1989; 1991). Dissanayake et al. (2001) infer robot positions by simultaneous localization and mapping (SLAM). However, maps are usually unavailable in unseen environments.

Recently, reinforcement learning (RL) has been applied in visual navigation. In general, it takes visual observations as inputs and predicts navigation actions directly. Mirowski et al. (2016) develop a navigation approach in 3D maze environments and introduce depth prediction and loop closure classification tasks to improve navigation performance. Parisotto & Salakhutdinov (2017) investigate a memory system to navigate in mazes. Some methods (Sepulveda et al., 2018; Chen et al., 2019; Savinov et al., 2018) use both visual features and the topological guidance of scenes for navigation, while natural-language instructions are employed to guide an agent to route among rooms (Anderson et al., 2018b; Wang et al., 2019; Deng et al., 2020; Hu et al., 2019; Majumdar et al., 2020; Hao et al., 2020). We notice that transformer architectures are also employed by Hao et al. (2020), named Prevalenet. However, Prevalenet is used to model languages and predict camera angles rather than encoding local and global visual features. Hence, Prevalenet is essentially different from our VT. Furthermore, Kahn et al. (2018) design a self-supervised approach to model environments by reinforcement learning. Tang et al. (2021) customize a specialized network for visual navigation via an Auto-Navigator. A Bayesian relational memory is introduced by Wu et al. (2019) to explore the spatial layout among rooms rather than steering an agent to desired objects with least steps. Meanwhile, Shen et al. (2019) employ multiple visual representations to generate multiple actions and then fuse those actions to produce an effective one. However, requesting such a large number of visual representations may restrict the transferring ability of a navigation system and increases the difficulty of data labeling. Note that Fang et al. (2019) propose a transformer to select the embedded scene memory slot, while our VT is designed to learn expressive visual representations correlated with directional signals.

Target-oriented visual navigation methods aim at steering an agent to object instances of a specified category in an unseen environment using least steps. Zhu et al. (2017) search a target object given in an image by employing RL to produce navigation actions based on visual observations. Mousavian et al. (2019) take semantic segmentation and detection masks as visual representations and also employ RL to learn navigation policies. Yang et al. (2018) exploit relationships among object categories for navigation, but they need an external knowledge database to construct such relationships. Wortsman et al. (2019) exploit word embedding (*i.e.*, GloVe embedding) to represent the target category and introduce a meta network mimicking a reward function for navigation. Furthermore, Du et al. (2020) introduce an object relation graph, dubbed ORG, to encode visual observations and design a tentative policy for deadlock avoidance during navigation. In ORG, object features are extracted from the second layer of the backbone in Faster R-CNN (Ren et al., 2015) and thus not the most prominent ones across the feature pyramid. Additionally, ORG chooses one instance with the highest confidence per category from detection results, and it may be affected by the false positive.

## 3 VISUAL NAVIGATION REVISIT

In this section, we mainly revisit the definition of object goal navigation and its general pipeline.

### 3.1 TASK DEFINITION AND SETUP

In this object goal visual navigation task, prior knowledge about the environment, *i.e.* topological map and 3D meshes, and additional sensors, *i.e.* depth cameras, are not available to an agent. RGB images in an egocentric view are the only available source to an agent, and the agent predicts its actions based on the current view and previous states. Following the works (Wortsman et al., 2019; Du et al., 2020), an environment is divided into grids and agents move between grid points via 6 different actions, consist of `MoveAhead`, `RotateLeft`, `RotateRight`, `LookUp`, `LookDown`, `Done`. To be specific, the forward step size is 0.25 meters, and the angles of turning-left/right and looking-up/down are 45° and 30°, respectively. An episode is defined as a success

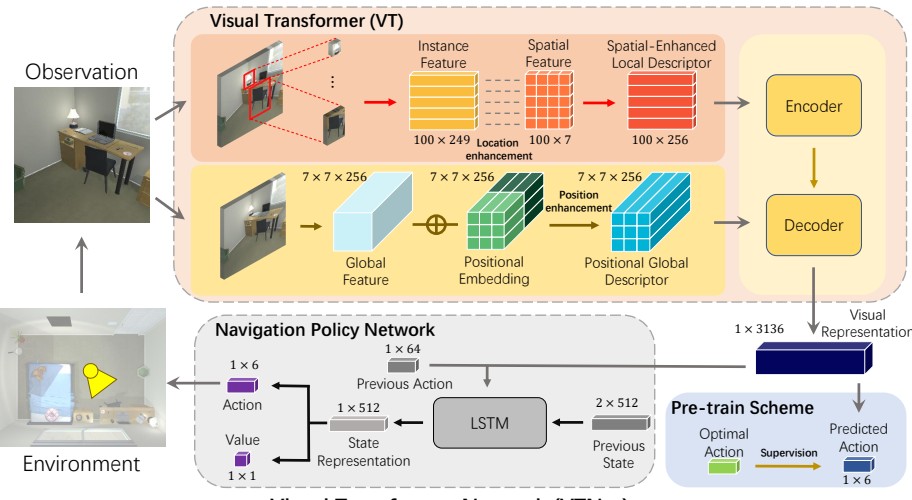

Figure 2: **Overview of our visual transformer navigation system.** Our visual transformer navigation network (VTNet) involves a visual transformer (VT) and a navigation policy network. The agent first fuses instance features and spatial features into spatial-enhanced local descriptor. Meanwhile, the positional global descriptor is obtained by adding a positional embedding to the global feature. Then the visual representation is decoded from these two spatial-aware descriptors by our VT. Our VTNet is pre-trained with the supervision of optimal navigation actions. The navigation policy network adopts A3C architecture and is trained with navigation rewards after pre-training.

when the following three requirements are met simultaneously: (i) the agent chooses the ending action Done within allowed steps; (ii) a target is in the view of the agent; (iii) the distance between the agent and the target is less than the threshold (*i.e.* 1.5 meters). Otherwise, the episode will be regarded as a failure.

A target class $T \in \{Sink, \ldots, Microwave\}$ and a start state $s = \{x, y, \theta_r, \theta_h\}$ are set randomly at the beginning of each episode, where $x$ and $y$ represent the coordinates, $\theta_r$ and $\theta_h$ indicate the view of a monocular camera. At each timestamp $t$, the agent records the observed RGB image $O_t$ from its monocular camera. Given the observation $O_t$ and the previous state $h_t$, the agent employs a visual navigation network to generate a policy $\pi(a_t|O_t, h_t)$, where $a_t$ represents the distribution of actions at time $t$. The agent selects the action with the highest probability for navigation.

## 3.2 PIPELINE

A typical pipeline of visual navigation consists of two parts, visual representation learning and navigation policy learning. **(i) Visual representation learning:** To encode the current observation in a compact way, existing works extract visual features from an image and then transform them into a vector-based representation, where direct concatenation (Wortsman et al., 2019) or graph embedding (Du et al., 2020) are used. **(ii) Navigation driven by visual features:** Once visual features are extracted, a navigation policy network that generates an action in each step for an agent will be learned. There are several ways to learn policy networks, such as Q-learning (Watkins & Dayan, 1992), PPO (Schulman et al., 2017) and A3C (Mnih et al., 2016). As navigation policy learning is not our focus, we adopt the standard Asynchronous Advantage Actor-Critic (A3C) architecture (Mnih et al., 2016). The navigation policy network takes the combination of the current visual representation, the previous action and state embedding as input, and outputs the action distribution and value. The agent selects actions with the highest probability from the predicted policy and uses the predicted value to train the navigation policy network.

## 4 PROPOSED VISUAL TRANSFORMER NETWORK

As illustrated in Figure 2, our visual navigation system includes two parts: (i) learning visual representations from RGB observations; (ii) learning navigation policy from the visual representations and previous states. In our VTNet, we introduce a visual transformer (VT) in the first part to ex-

plore the relationship among *all objects* and their spatial correlations. In VT, we further design two spatial-aware descriptors, *i.e.*, a spatial-enhanced local descriptor and a positional global descriptor, to allow us extract visual information effectively. Then, our VT fuses these two types of descriptors with a multi-head attention operation to produce final visual representations. Moreover, our VT enforces visual representations to be highly correlated to navigation signals via our developed pre-training scheme, thus easing the training difficulty of VT and facilitating navigation policy learning.

### 4.1 SPATIAL-ENHANCED LOCAL DESCRIPTOR

To learn the relationship among all the instances, we first perform object detection and locate all the object instances of interest by a detector DETR (Carion et al., 2020). DETR transforms $N$ encoded $d$-dimension features $\mathbb{R}^{N \times d}$ from the same layer to $N$ detection results, including the bounding boxes, confidence and semantic labels by a feed forward network. Note that ORG (Du et al., 2020) extracts object features from the second layer of the backbone in Faster R-CNN based on the predicted bounding-boxes rather than the penultimate layer of the classifier in Faster R-CNN as in the work (Anderson et al., 2018a). Hence, ORG features are not the most prominent ones across the feature pyramid and scale-sensitive. In contrast, features extracted by DETR not only contain bounding-boxes and class labels but also are scale-robust as features are aligned by DETR decoder, *i.e.*, output from the penultimate layer.

*Remark.* Benefiting from our VT, we leverage all the information of the detected objects while Du et al. (2020) only select the proposal with the highest confidence in each category. Therefore, the agents in ORG will miss important information from other objects of the same class or might be severely affected if selected proposals are false positive. In contrast, our VT preserves all the information, and thus our agents are able to exploit the relationship among instances. This makes our visual representation more comprehensive and essentially different from prior works.

Our local spatial feature is obtained by concatenating the normalized bounding box, confidence and top-rated semantic label for each object. To indicate the target class to an agent, we also concatenate a one-hot encoded target vector $\mathbb{R}^{N \times 1}$ with our spatial feature $\mathbb{R}^{N \times 7}$. After obtaining the instance feature and spatial feature, we employ a multi-layer perceptron (MLP) (*i.e.*, two fully-connected layers with ReLU) and fuse them to a spatial-enhanced local descriptor $L \in \mathbb{R}^{N \times d}$ so as to act as the key of our VT encoder.

### 4.2 POSITIONAL GLOBAL DESCRIPTOR

In addition to the spatial-enhanced local descriptor, agents require a global feature to describe the surrounding environment. Similar to SAVN (Wortsman et al., 2019), we adopt ResNet18 (He et al., 2016) pretrained on ImageNet (Deng et al., 2009) to extract global features of the observations. Given a global feature $\mathbb{R}^{h \times w \times D}$, we first employ $1 \times 1$ convolution to reduce the channel dimension of a high-level activation map from $D$ to a smaller dimension $d$, where $h$ and $w$ represent the height and width of activation maps, respectively. This ensures that global descriptors have the same dimension as the key of our VT.

Unlike previous works that directly concatenate a global feature as a part of the visual representation, we introduce a positional global descriptor as the query in our VT decoder. A region feature only represents visual contents in each region. To emphasize the region position information, we incorporate a positional embedding to each region feature. Then we add positional encoding $\mathbb{R}^{h \times w \times d}$ to the global feature. Let $u$ and $v$ represent the row and column indexes of an image region respectively, and $i$ is the index along the dimension $d$. Our positional embedding is expressed as:

$$PE_{2i}(u,v) = \begin{cases} \sin(\frac{u}{10000^{2i/d}}), & 0 < i \leq \frac{d}{2} \\ \sin(\frac{v}{10000^{2i/d}}), & \frac{d}{2} < i \leq d \end{cases} \quad PE_{2i+1}(u,v) = \begin{cases} \cos(\frac{u}{10000^{2i/d}}), & 0 < i \leq \frac{d}{2} \\ \cos(\frac{v}{10000^{2i/d}}). & \frac{d}{2} < i \leq d \end{cases} \quad (1)$$

Therefore, each global feature represents one particular region of the observation. Finally, we reshape positional embedded global features into a matrix-based representation, namely positional global descriptor $G \in \mathbb{R}^{hw \times d}$.

### 4.3 VISUAL TRANSFORMER

After obtaining our extracted spatial-enhanced and positional global descriptors, we introduce our visual transformer.

**Encoder.** In order to exploit the spatial relationship between detected instances and observed regions, we attend spatial-enhanced local descriptors to positional global descriptors via a transformer. We first feed the spatial-enhanced local descriptors into the encoder as keys and values by employing multi-head self-attention. Following the transformer architecture (Vaswani et al., 2017; Fan et al., 2021), each encoder layer consists of a multi-head self-attention module and a feed-forward layer.

**Decoder.** Inspired by human navigation behaviors, we aim to explore the correspondences between observation regions and navigation actions. For example, once an agent notices a target lying on the right side of the field of view, it should prioritize to select `RotateRight` instead of `RotateLeft`. Since each positional global descriptor corresponds to a certain region of the observation, we refer to the positional global descriptor as the location query and feed the query into the decoder. Given positional global descriptor $G$ and encoded spatial-enhanced local descriptor $L'$, our attention function of visual transformer decoder is expressed as:

$$Attention(G, L') = softmax(\frac{GL'^T}{\sqrt{d}}L').$$ 

(2)

### 4.4 PRE-TRAINING VISUAL TRANSFORMER

We observed that directly feeding the decoded representation from our VT to a navigation network, we fail to learn successful navigation policy. This is mainly because training a deep VT is very difficult especially when the supervision signals are provided by a weak reward from reinforcement learning. Therefore, the decoded features might be uninformative and confuse an agent. The agent would prefer to choose the termination action (often around 5 steps in our experiments) in order to reduce penalties from reinforcement learning.

To address the aforementioned issue, we propose a pre-training scheme for our VT. To be specific, we enforce the decoded features to be expressive by introducing an imitation learning task, as seen in Figure 2. Concretely, human navigation behaviors can be predicted from the decoded representations in a step-wise fashion. We use Dijkstra's Shortest Path First algorithm to generate optimal action instructions as human expert experience. Under the supervision of optimal action instructions, our VT learns to imitate the optimal navigation action selection.

In the pre-training stage, we do not employ our navigation network (*i.e.*, LSTM), and previous actions as well as states are not available. Note that, in our navigation network, previous actions, previous states and current visual representations are exploited, as seen in Figure 2. Thus, we replace our LSTM with an MLP and predict action distributions based on the current visual representation. A cross-entropy loss $L_{vt} = CE(a_t, \hat{a})$ is employed to train our VT and the MLP, where $a_t$ is the predicted action, $\hat{a}$ represents the optimal action instruction and $CE$ indicates the cross-entropy function. After pre-training, features from our VT also exhibit strong association with directional navigation signals as only an MLP is employed on top of the features. Therefore, the decoded features will facilitate the navigation network training.

## 5 EXPERIMENTS

### 5.1 PROTOCOLS AND EXPERIMENTAL DETAILS

**Dataset.** We perform our experiments on AI2-Thor (Kolve et al., 2017), an artificial 3D environment with realistic photos. It contains 4 types of scenes, *i.e.*, kitchen, living room, bedroom and bathroom. In each type of scenes, there are 30 different rooms with various furniture placements and items. Following Du et al. (2020), we choose 22 categories as the target classes and ensure that there are at least 4 potential targets in each room.

We use the same training and evaluation protocols as the works (Wortsman et al., 2019; Du et al., 2020). 80 rooms out of 120 are selected as the training set while each scene contains 20 rooms.

Table 1: Comparison with the state-of-the-art. We report the average success rate (%) and SPL as well as their variances in parentheses by repeating experiments five times. $L > 5$ represents the episodes which require at least 5 steps.

| Method | ALL | | $L \geq 5$ | |
|---|---|---|---|---|
| | Success | SPL | Success | SPL |
| Random | 8.0 (1.3) | 0.036 (0.006) | 0.3 (0.1) | 0.001 (0.001) |
| WE | 33.0 (3.5) | 0.147 (0.018) | 21.4 (3.0) | 0.117 (0.019) |
| SP (Yang et al., 2018) | 35.1 (1.3) | 0.155 (0.011) | 22.2 (2.7) | 0.114 (0.016) |
| SAVN (Wortsman et al., 2019) | 40.8 (1.2) | 0.161 (0.005) | 28.7 (1.5) | 0.139 (0.005) |
| ORG (Du et al., 2020) | 65.3 (0.7) | 0.375 (0.008) | 54.8 (1.0) | 0.361 (0.009) |
| ORG+TPN (Du et al., 2020) | 69.3 (1.2) | 0.394 (0.010) | 60.7 (1.3) | 0.386 (0.011) |
| Baseline | 62.6 (0.9) | 0.364 (0.006) | 51.5 (1.2) | 0.345 (0.007) |
| VTNet | 72.2 (1.0) | **0.449** (0.007) | 63.4 (1.1) | **0.440** (0.009) |
| **VTNet + TPN** (Du et al., 2020) | **73.5** (1.3) | 0.440 (0.009) | **63.9** (1.5) | **0.440** (0.011) |

We equally divide the remaining 40 rooms into validation and test sets. We report the results of the testing data by using the model with the highest success rate on the validation set.

**Evaluation metrics.** We evaluate our model performance by success rate and Success Weighted by Path Length (SPL). The success rate measures navigation effectiveness and is computed by $\frac{1}{N} \sum_{n=0}^{N} S_n$, where $N$ is the number of episodes and $S_n$ is a success indicator of the $n$-th episode. We adopt SPL to measure the navigation efficiency. Given the length of the $n$-th episode $Len_n$ and its optimal path $Len_{opt}$, SPL is formulated as $\frac{1}{N} \sum_{n=0}^{N} S_n \frac{Len_n}{max(Len_n, Len_{opt})}$.

**Training details.** We use a two-stage training strategy. In Stage 1, we train our visual transformer for 20 epochs with the supervision of optimal action instructions. In this fashion, we explicitly construct the association between visual representations and navigation actions. In Stage 2, we train the navigation policy for 6M episodes in total with 16 asynchronous agents. We set a penalization $-0.001$ on each action step and a large reward 5 when an agent completes an episode successfully. We adopt DETR as the object detector and fine-tune DETR on the AI2-Thor training dataset. In training DETR, we applied data augmentation, such as resize and random crop. We use the Adam optimizer (Kingma & Ba, 2014) to update the policy network with a learning rate $10^{-4}$ and the pre-trained VT with a learning rate $10^{-5}$. Our codes and pre-trained model will be publicly released for reproducibility.

## 5.2 COMPETING METHODS

We compare our method with the following ones: **Random policy.** An agent chooses actions based on a uniform action probability. Thus, the agent will walk or stop in a scene randomly. **Scene Prior (SP)** (Yang et al., 2018) learns a graph neural network from the FastText database (Joulin et al., 2016) and leverages the scene prior knowledge and category relationships for navigation. **Word Embedding (WE)** uses GloVe embedding (Pennington et al., 2014) to indicate the target category rather than detection. The association between object appearances and GloVe embeddings is learned through trail and error. **Self-adaptive Visual Navigation (SAVN)** (Wortsman et al., 2019) introduces a meta reinforcement learning method that allows an agent to adapt to unseen environments. **Object Relationship Graph (ORG)** (Du et al., 2020) is a visual representation learning method to encode correlation among categories and employs a tentative policy network (TPN) to escape from deadlocks. **Baseline** is a vanilla version of VTNet. We feed the concatenation of the local instance features from DETR and the global feature to A3C for navigation. Note that, our baseline does not employ spatial-enhanced local and positional global descriptors as well as our visual transformer.

## 5.3 EVALUATION RESULTS

**Improvement over Baseline.** Table 1 indicates that VTNet surpasses the baseline by a large margin on both success rate (+9.6%) and SPL (+0.085). **Baseline** only resorts to the detection features and global feature for navigation. The relations among local instances and the association between the visual observations and actions are not exploited. This comparison suggests that our VT leads to informative visual representations for navigation, and thus significantly improves the effectiveness and efficiency of our navigation system.

**Comparison with competing methods.** As indicated in Table 1, we observe that VTNet significantly outperforms SP (Yang et al., 2018) and SAVN (Wortsman et al., 2019). Since SP and SAVN

Figure 3: **Visual results of four different models in testing environments.** The target objects (*i.e.*, *RemoteControl*) are highlighted by the blue boxes. Green and red curves represent success and failure cases, respectively. The episode produced by our VTNet is successful in reaching the target and use shortest steps. In comparison, ORG takes more steps to reach the target. SAVN and Baseline miss both targets.

employ word embedding as a target indicator while VTNet replaces word embedding with our VT, our method achieves expressive object and image region representations for navigation. In addition, SP and SAVN concatenate features from various modalities directly to generate visual representations. The gap between different modalities may not facilitate navigation policy learning. In contrast, benefiting from our pre-training, features from our VT are more correlated to navigation actions, thus expediting navigation policy learning.

Our method outperforms the state-of-the-art method ORG (Du et al., 2020) by +2.9% in success rate and +0.055 in SPL. Moreover, when ORG does not employ TPN, the advantage of our method becomes more obvious (6.9% improvement), and this mainly comes from our superior visual presentations. Since ORG only chooses an object with the highest confidence in each class, the relationship among objects is not comprehensive. On the contrary, our method can exploit all the detected instances to deduce the relationships among objects due to our VT architecture. Moreover, since DETR infers the relations between object instances and the global image context, the local features output by the DETR are more informative compared to the object features used in ORG. This can be proved by the result when we use Faster R-CNN as our backbone, as indicated by Table 2. We also show a case study in Figure 3 (more visual results are provided in the appendix). Furthermore, we also try to employ TPN to improve our navigation policy. As seen in Table 1, VTNet+TPN improves the success rates but the improvement is not as much as ORG+TPN. This also implies that our visual representations significantly facilitate navigation action selections.

**Case Study.** As illustrated in Figure 3, SAVN and Baseline both issue the termination command after navigating a few steps (7 and 19 steps, respectively), but fail to reach the target. This indicates that the relationships among categories are not clear in SAVN and Baseline. In contrast, both ORG and VTNet find the target. Since our visual transformer provides clear directional signals, VTNet uses the least steps to find the object.

## 5.4 Variant and Ablation Study

In this section, we analyze the impact of each component in VTNet, including the spatial-enhanced local descriptor, positional global descriptor, visual transformer that fuses these two spatial-aware descriptors and pre-training scheme.

To illustrate the necessity of the spatial enhancement, we directly use the object features without spatial enhancement. In this case, our network fails to converge because the feed-forward layers that predict bounding-boxes and class labels in DETR are not used in VTNet and spatial information cannot be decoded by the navigation network. Thus, spatial enhancement allows an agent to exploit instance location information explicitly.

As indicated in Table 2, we achieve better navigation performance using instance features from DETR compared to employing Faster R-CNN features following the feature extraction of Du et al. (2020) ("Faster R-CNN"). Unlike Faster R-CNN, DETR infers the relations between object instances and the global image context via its transformer to output the final predictions (*i.e.*, class labels and bounding boxes). Although DETR and Faster R-CNN achieve similar detection performance (Carion et al., 2020), features extracted by DETR are more informative and robust than those of Faster R-CNN used in ORG. Specifically, ORG extracts features from the second layer of the backbone in Faster R-CNN based on the predicted bounding-boxes to ensure the features are comparable, but the features are not the most prominent ones across the feature pyramid. Therefore, the

Table 2: Impacts of different components on navigation performances. Faster R-CNN and Faster R-CNN[†] represent instance features extracted by Faster R-CNN following Du et al. (2020) and Anderson et al. (2018a), respectively.

| Method | | w/o global | w/o decoder | w/o pe | VTNet$_g$ | Baseline | | | VTNet | | |
|---|---|---|---|---|---|---|---|---|---|---|---|
| | | | | | | Faster R-CNN | Faster R-CNN[†] | DETR | Faster R-CNN | Faster R-CNN[†] | DETR |
| ALL | Success | 67.0 (2.8) | 67.0 (1.4) | 71.0 (0.7) | 70.1 (1.3) | 56.4 (0.9) | 57.2 (1.1) | 62.6 (0.8) | 70.1 (1.0) | 70.3 (1.2) | **72.2** **(1.0)** |
| | SPL | 0.390 (0.021) | 0.373 (0.013) | 0.432 (0.009) | 0.411 (0.009) | 0.319 (0.007) | 0.308 (0.008) | 0.365 (0.010) | 0.396 (0.010) | 0.387 (0.012) | **0.449** **(0.007)** |
| $L \geq 5$ | Success | 54.5 (3.1) | 53.2 (1.6) | 61.2 (0.9) | 60.6 (1.5) | 42.5 (1.2) | 46.7 (1.3) | 51.5 (1.0) | 61.7 (1.2) | 62.1 (1.4) | **63.4** **(1.1)** |
| | SPL | 0.357 (0.021) | 0.343 (0.017) | 0.416 (0.010) | 0.395 (0.011) | 0.270 (0.009) | 0.276 (0.010) | 0.345 (0.012) | 0.399 (0.016) | 0.376 (0.012) | **0.440** **(0.009)** |

object features "Faster R-CNN" extracted by ORG are inferior to DETR features, and our navigator employing DETR features outperforms ORG.

Moreover, we adopt the instance features from Faster R-CNN following the feature extraction fashion of Anderson et al. (2018a) ("Faster R-CNN[†]"). Thus, we obtain the instance features from the penultimate layer of the classifier in Faster R-CNN. We observe that instance features from DETR also improve the navigation performance compared to the Faster R-CNN[†] features. We speculate the improvements mainly come from the fact that the features output by DETR decoder have embedded global context information, and those features are more suitable for the feature fusion operations.

As seen in Table 2, we first remove the global feature from our system ("VTNet w/o global"), and the navigation performance degrades significantly. This validates the importance of global features, which provide contextual guidance to an agent. Moreover, when we remove the positional embedding from the global feature ("VTNet w/o pe"), we observe that both effectiveness and efficiency of the navigation decrease. This indicates that the position embeddings facilitate our VT to exploit the spatial information of observation regions. Furthermore, when we remove the VT decoder ("VTNet w/o decoder") and concatenate the global and local descriptors directly, our method suffers performance degradation. This demonstrates that our VT plays a critical role in attending the global descriptors to local ones. Additionally, we feed the positional global features into the transformer encoder ("VTNet$_g$"). The navigation performance of VTNet$_g$ is superior to that of ORG but slightly inferior to the performance of our VTNet. This demonstrates the transformer architecture is effective to extract informative visual representations, and assigning different functions to different modules would further facilitate the establishment of mappings in our VT.

When the pre-training scheme is not applied to our VT, our agent fails to learn any effective navigation policy and thus we do not report the performance. This manifests that our VT pre-training procedure provides a good initialization to our transformer and prior knowledge on associating visual observations with navigation actions to agents.

## 6 CONCLUSION

In this paper, we proposed a powerful visual representation learning method for visual navigation, named Visual Transformer Network (VTNet). In our VTNet, a visual transformer (VT) has been developed to encode visual observations. Our VT leverages two newly designed spatial-aware descriptors, *i.e.*, a spatial-enhanced local object descriptor and a positional global descriptor, and then fuses those two types of descriptors via multi-head attention to achieve our final visual representation. Thanks to our VT architecture, all the detected instances will be exploited for understanding the current observation. Therefore, our visual representation is more informative compared to that used in state-of-the-art navigation methods. Benefiting from our pre-training strategy, our VT is able to associate visual representations with navigation actions, thus significantly expediting navigation policy learning. Extensive results demonstrate that our VTNet outperforms the state-of-the-art in terms of effectiveness and efficiency.

## ACKNOWLEDGMENTS

This work was supported by the ARC Discovery Early Career Researcher Award (DE200101283), the ARC Discovery Project (DP210102801) and the Data61 Collaborative Research Project.

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

# A APPENDIX

## A.1 FEATURE DETAILS IN VTNET

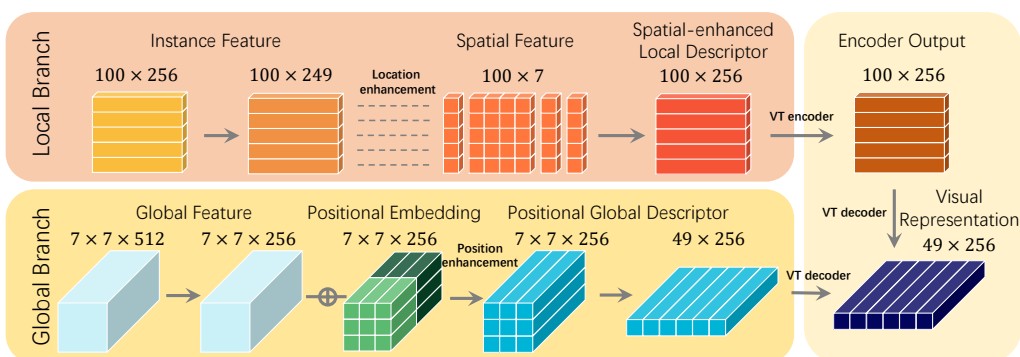

Figure 4: Illustration feature flowchart in VTNet.

For reproducibility, we illustrate the detailed feature flowchart of our VTNet in Figure 4. We extract instance features $\mathbb{R}^{100\times256}$ and location features $\mathbb{R}^{100\times249}$, and concatenate them into a spatial-enhanced local descriptor $\mathbb{R}^{100\times256}$. The local branch integrates semantic labels $\mathbb{R}^{100\times1}$, bounding boxes $\mathbb{R}^{100\times4}$, confidences $\mathbb{R}^{100\times1}$ and the target labels $\mathbb{R}^{100\times1}$ for the current observation. In the global branch, the positional embedding $\mathbb{R}^{7\times7\times256}$ is added to the global feature $\mathbb{R}^{7\times7\times256}$, leading to a positional global descriptor $\mathbb{R}^{49\times256}$. The spatial-enhanced local and positional global descriptors are fused by VT encoder and then the visual representation $\mathbb{R}^{49\times256}$ is output by our VT decoder.

## A.2 FAILURE CASE STUDY

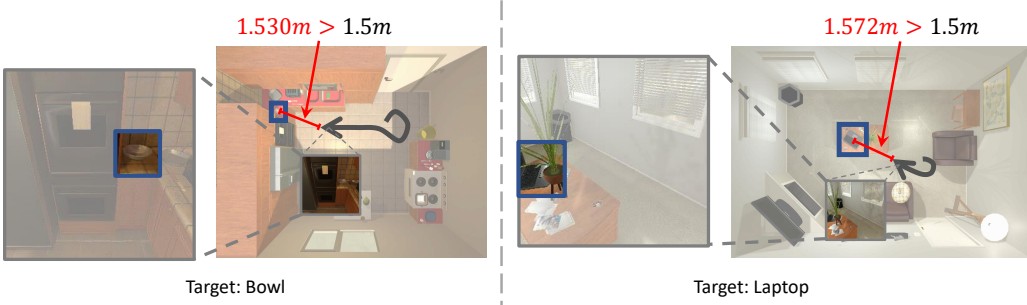

Figure 5: **Visual results of failure cases in testing environments.** The target objects (*i.e.*, Bowl and Laptop) are highlighted by the blue boxes. Red lines indicate the distance between the agent and the target object. Gray curves represent trajectories of agents. Both episodes fail because distances (*i.e.*, 1.530m and 1.572m) between the agent and the target in the field of view are larger than the threshold (*i.e.*, 1.5m).

As demonstrated in Figure 5, our VTNet fails to reach targets because the distances between agents and targets are larger than the threshold distance (*i.e.*, 1.5m). In these two failure cases, agents find targets but implement the termination action before reaching a position closer than the threshold. Due to the lack of depth information and variances of target object sizes, an agent may predict that it is within a 1.5 meter radius of the target by mistake and thus terminates current episode.

## A.3    CASE STUDY

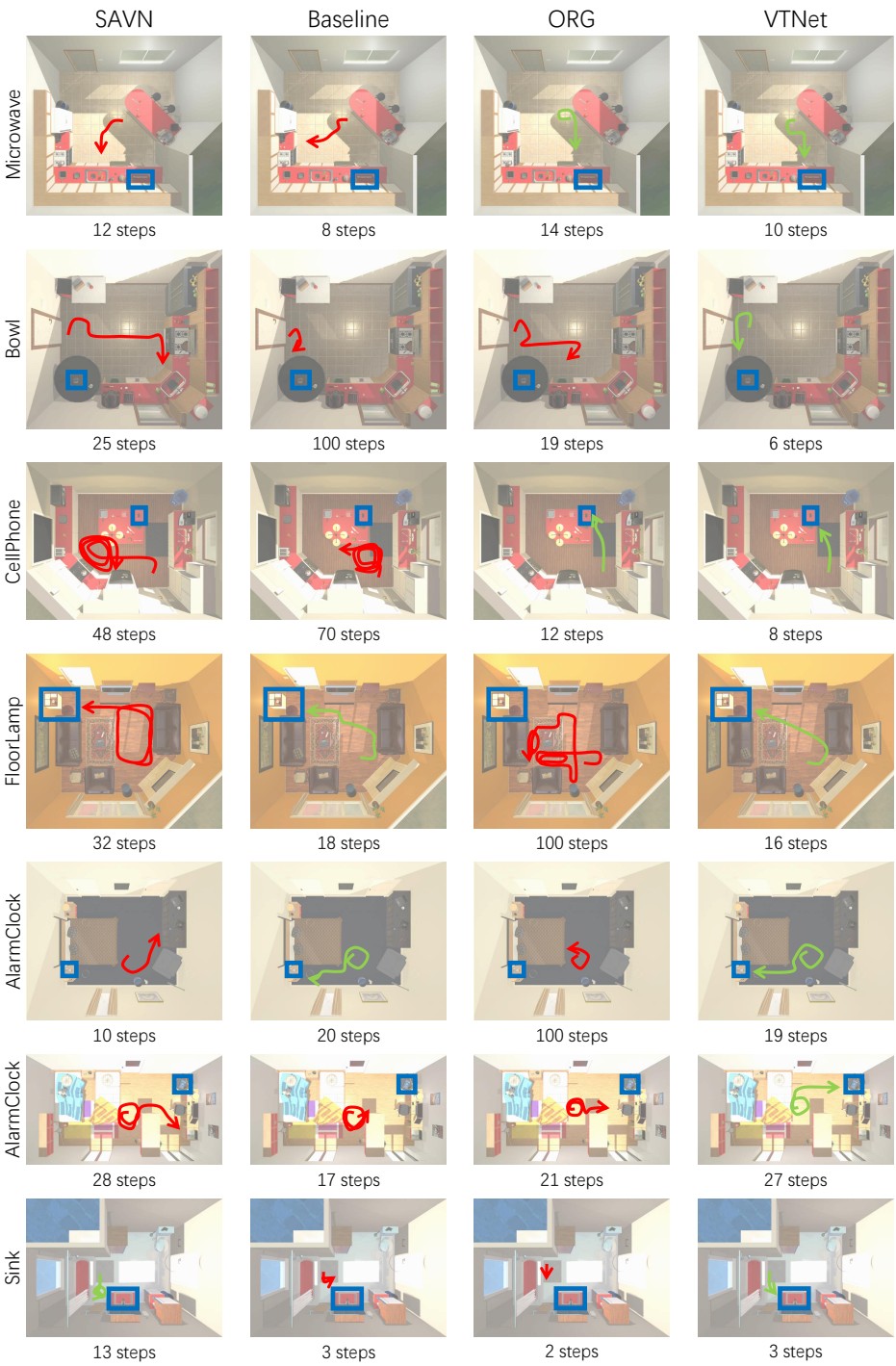

Figure 6: **Visual results of four different models in testing environments.** We compare VTNet with SAVN (Wortsman et al., 2019), Baseline and ORG (Du et al., 2020). The target objects are highlighted by the blue boxes. Green and red curves indicate success and failure cases, respectively. Our VTNet successfully reaches targets and uses the shortest steps.

## A.4 DIFFERENT VISUAL TRANSFORMER ARCHITECTURES.

Table 3: Comparison of visual transformer architectures. We report the pre-training accuracy on the validation dataset, navigation success rate and SPL on the test dataset.

| Multi-head numbers | Encoder layers | Decoder layers | Accuracy | Success | SPL |
|---|---|---|---|---|---|
| | 1 | 1 | 0.722 | 71.2 | 0.433 |
| **4** | **2** | **2** | **0.723** | **72.2** | **0.449** |
| | 4 | 4 | 0.715 | 70.0 | 0.419 |
| | 1 | 1 | 0.707 | 70.4 | 0.422 |
| 8 | 2 | 2 | 0.718 | 70.9 | 0.436 |
| | 4 | 4 | 0.710 | 68.9 | 0.423 |
| | 6 | 6 | 0.701 | 68.2 | 0.411 |

We construct different transformer architectures by varying the number of encoder and decoder layers. Table 3 summarizes the performance of these architectures. We observe that as a visual transformer becomes too deep, a transformer may fail to converge to an optimal policy. On the other hand, a transformer with a single encoder and decoder layer does not have sufficient network capability to produce representative features. The highest success rate is achieved when a VT contains four multi-head self-attention mechanism modules and two layers in the encoder and decoder.

## A.5 NECESSITY OF PRE-TRAINING SCHEME

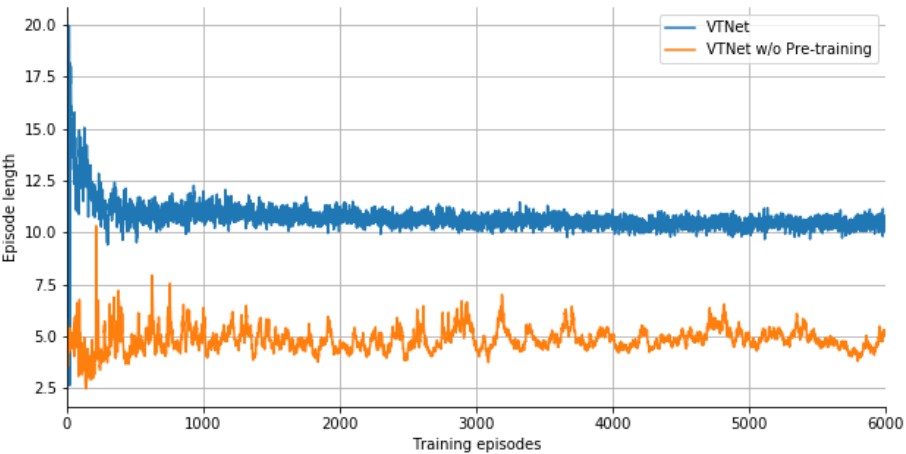

Figure 7: **Average episode lengths of VTNet and VTNet without pre-training during training.** We compare VTNet with VTNet without pre-training scheme. Blue and orange curves represent VTNet and VTNet w/o pre-training, respectively.

As demonstrated in Figure 7, our VTNet spends nearly 10 steps per episode in training, while the navigator w/o pre-training scheme often fails to reach targets and stops around 5 steps after being trained tens of thousands of episodes. Due to the large parameters and complex architectures of transformers, it is often difficult to train our transformers from scratch (Liu et al., 2020). Without a good initialization for our VT, it is very difficult to learn our VT and policy network in an end-to-end fashion with RL rewards. This is because the visual representations from VT are not informative or even meaningless and the inferior visual representations would harm policy network learning. As a result, the navigation policy network may be trapped into a local minimum (i.e., terminating navigation early to avoid more penalties) and our VT cannot receive positive rewards from preceding trajectories.

## A.6 ADDITIONAL VISUALIZATION RESULTS

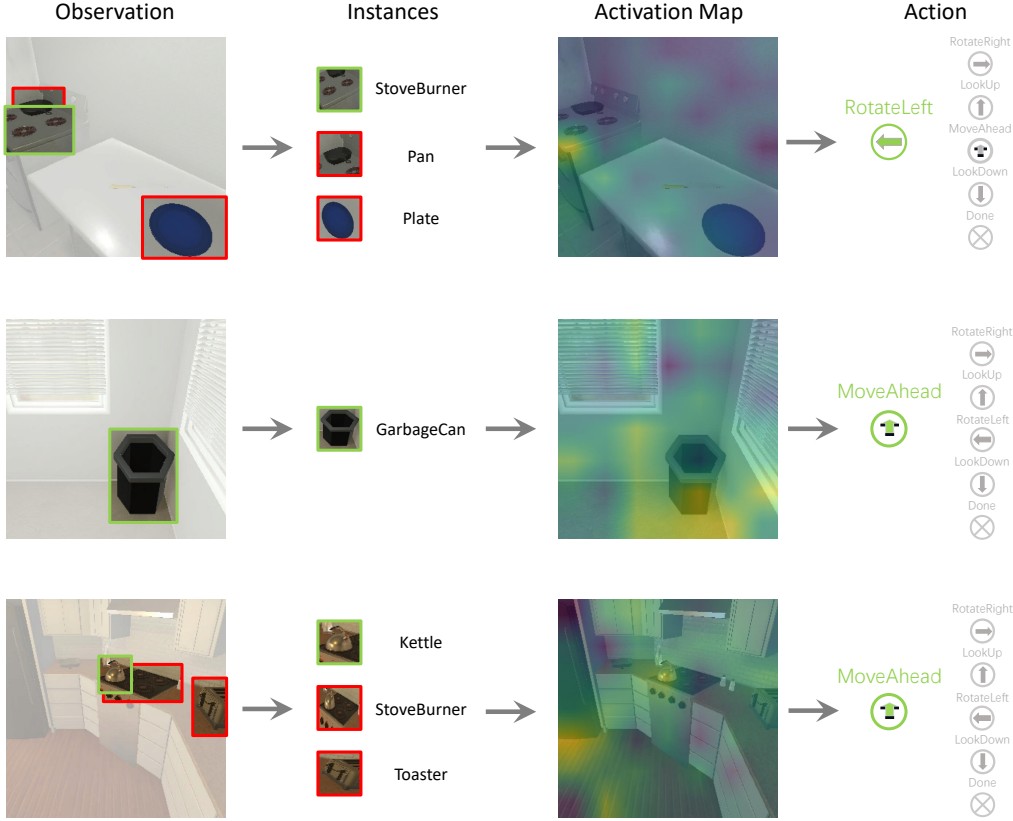

Figure 8: **Visualizations of attention scores.** The target classes (*i.e.*, StoveBurner, GarbageCan, Kettle) are highlighted by green bounding boxes. Our agent detects the instances of interest and then attends the detected instances to the global image regions by our VT. We observe that high attention scores are obtained on the areas corresponding to the targets. Guided by the visual representations, the agent selects actions to approach the targets.

