# OpenReview forum: "VTNet: Visual Transformer Network for Object Goal Navigation"
_ICLR.cc/2021/Conference — ICLR 2021 Poster_

### Official Review · AnonReviewer4 · 2020-10-18
**Effective method, good results, but writing requires improvement**

**Rating:** 6
**Confidence:** 4

**Review:**

**Paper summary**

The paper addresses the problem of navigation towards objects (ObjectNav) in a virtual environment. The idea of the paper is to incorporate spatial information of objects using a transformer-based framework called Visual Transformer Network. The paper compares the results with a number of state-of-the-art ObjectNav models and provides an ablation study. The results have been reported on the AI2-THOR framework.

**Paper strengths**

- The idea of incorporating object information using transformers for a navigation agent is new.

- The proposed method outperforms a number of strong baselines.

- The ablation studies show that the introduced components are effective.

**Paper weaknesses**

- It is hard to understand some parts of the paper. For example, the introduction discusses details such the difference between DETR and Faster RCNN or difficulty of training the transformers. It is difficult to understand these details without knowing the proposed method. The introduction should provide a high-level overview of the paper instead of these types of details. Also, the paper requires proof reading. There are several sentences with grammar issues.

- It is a bit strange that nothing is learned without the imitation pre-training. It would be good to dig deeper and provide a better explanation for why this happens.

- Equation 1 is not clear. A brief explanation would help.

- I recommend running the method on some other frameworks which include slightly larger scenes to see if the method generalizes to those as well. RoboTHOR (https://github.com/allenai/robothor-challenge) is very close to the framework used in this paper so it might be a good choice for these experiments.

**Justification of rating**

Overall, I am leaning towards accepting this paper since it introduces a new way for incorporating object information and it outperforms strong object navigation baselines. Writing is the main issue of this paper.

**Post-rebuttal**

I read the rebuttal and the other reviews. The rebuttal addresses my concerns to some extent (writing has improved in the revised version, but it still has some issues). So I am going to keep my rating.

---

> ### Author Response · Authors · 2020-11-24
> **Providing high-level overview of VTNet and explanations on pre-training necessity**
>
> We Thank R4 for the constructive comments and the acknowledgment of our novelty and contributions. As suggested, we have carefully revised our manuscript.
>
>
> **Q1: High-level overview in the introduction**
>
> We introduce our method in the first paragraph of our introduction. Our navigation system consists of three stages: (i) we first extract the local object descriptors and a global descriptor from the current RGB image; (ii) we then fuse these two descriptors to achieve the visual representations by presenting a visual transformer (VT); (iii) we pre-train our VT and then train the entire navigation network in an end-to-end fashion.
> We will reorganize the introduction and correct the grammar issues in the revised version.
>
>
> **Q2: Necessity of pre-training scheme**
>
> Thank you for the suggestion. Due to the large parameters and complex architectures of transformers, it is often hard to train our transformers from scratch (Liu et al., 2020). Without a good initialization for our VT, it is very difficult to learn our VT and policy network in an end-to-end fashion with RL rewards. This is because the visual representations from randomly initialized VT are not informative or even meaningless and the inferior visual representations would harm policy network learning. As a result, the navigation policy network may be trapped into a local minimum (i.e., terminating navigation early to avoid more penalties) and our VT cannot receive positive rewards from preceding trajectories. As seen in Figure 7, our navigator w/o pre-training scheme stops around 5 steps after training tens of thousands of episodes.
>
> To overcome this issue, we introduce our pre-training scheme. After warming-up, our VT embeds directional signals into visual representations, thus achieving a good initialization. Then, the positive rewards can lead our navigation system to obtain an optimal navigation policy.
>
>
> **Q3: Equation (1) is not clear**
>
> Thank you for pointing this out. We notice that there are typos in the equation which leads to confusion. The correct form is expressed as:
>
> $$ PE_{2i} (u, v) = \sin(\textit{u} / 10000^{2i / d}),  0 < i \leq d / 2 $$
> $$ ~ ~ ~ ~ ~ ~ ~ ~ ~ ~ ~ ~ ~ ~ ~ ~ ~ ~ ~ = \sin(\textit{v} / 10000^{2i / d}),  d / 2 < i \leq d $$
>
> $$ PE_{2i+1} (u, v) = \cos(\textit{u} / 10000^{2i / d}),  0 < i \leq d / 2 $$
> $$ ~ ~ ~ ~ ~ ~ ~ ~ ~ ~ ~ ~ ~ ~ ~ ~ ~ ~ ~ ~ ~ =  \cos(\textit{v} / 10000^{2i / d}), d / 2 < i \leq d $$
>
>
> where $u$ and $v$ represent the row and column indexes of an image region respectively, and $i$ is the index along the dimension $d$. We will correct the typos in the revision.
>
>
> **Q4: Experiments on RoboTHOR**
>
> We thank R4 for the constructive comments. Due to the timeframe of the rebuttal, we cannot finish training our model on RoboTHOR. We will evaluate our VTNet in the RoboTHOR environment as our future work.
>
> **References**
>
> [1] Liu, L., Liu, X., Gao, J., Chen, W., & Han, J. (2020). Understanding the Difficulty of Training Transformers. arXiv preprint arXiv:2004.08249.

---

### Official Review · AnonReviewer2 · 2020-10-28
**applied transformer on visual navigation tasks; good performance achieved on AI2-THOR**

**Rating:** 6
**Confidence:** 3

**Review:**

### summary
This paper introduces transformer network to visual navigation, specifically, object-goal navigation. It also develops several new feature descriptors as the input of the transformer encoder and decoder. To properly train the whole model, a supervised pre-training stage is used to warm-up the transformer model. Great performance has been achieved on AI-THOR benchmark.

### pros
1. Lots of people must have thought to use transformer to replace the RNN/LSTM in lots of visual navigation framework. This paper provides a good example. Most importantly, this paper focus on the representation learning part of the whole pipeline, which isn't that straightforward of how to use a transformer.
2. The writing is mostly clear with clear motivation and background discussion.
3. The performance boost, especially SPL, is relatively significant compared to previous SOTA, and the ablation studies have verified most of the design choices.

### cons
There are a couple of things which are not clear to me, or confused me when I was reading the paper:
1.  The writing in the approach section isn't very clear. First, it would be much better to define clear notations for all the features/descriptors, and use such notations in the figure. The current writing uses "instance feature", "global feature", "positional-global", "spatial feature", "spatial-enhanced"..., which are a little bit confusing to me. Second, I think most details are properly ignored in Fig.2. It becomes not as informative as the detailed version (Fig.4) in Appendix. Note that these two figures are not consistent that the "add" symbol for positional enhancement is missing is Fig.4. I also suggest that the positional embedding blob not crossing the arrow of global feature, they are just added together.Third, Sec.4.2 writes "we first reduce the channel dimension of a high-level activation map from D to a smaller dimension d", how the reduction is done exactly? From appendix it seems like a 256-dim vector is transformed into 249-dim. Fourth, $h$ and "w" are abused. In figure, they are annotated  on the long side of the tensor, in eq (1) they seem to be the output of positional embedding, and in Sec.4.2 description they are the resolution of 7. Similarly, $L$ is abused as it means input of encoder in Sec.4.1 but output of encoder in Sec.4.3. Let me stop here, but these things make the approach not super clear to me.
2. In Sec.4.1, I'm not fully convinced of the statement of faster rcnn even thought the experiments empirically verified it. Faster RCNN w/o FPN only output features after "conv4+ROI-pooling" (ResNet-101-C4 variant). Why is it blamed for scale-sensitive? Actually, what does scale-sensitive mean here? Why DETR doesn't suffer from it? Honestly I don't think that's the reason why Faster RCNN performs worse.
3. Also, I'm not fully convinced of the statement of the "early stopping" in Sec.4.4. The penalties are the same for different model in RL,  why this transformer based representation learner suffers from "early stopping"? Is there a plausible explanation? It's fine that you cannot conclude something for sure because transformers are always hard to train, but the statement in paper reads not super convincing to me.
4. Sec.5.1 SPL formulation seems to be wrong? The success indicator seems missing? The current equation is simply a ratio between any episode length over the optimal length regardless whether it's an success episode or not.
5. Why not also adding global features into the transformer encoder? For example, reshape and concat with the input. Is the encoder supposed to be local?

### misc
1. The best results of VTNet in Tab.1 used TPN. It might be better to introduce TPN in Appendix for completeness.
2. Variance is not reported in Tab.1, which is uncommon for RL/control paper.
3. Because transformer has attention module and the relationship can be easily visualized. I was expecting more interpretation/visualization like Fig.1 right to show the proposed methods actually attend to proper areas. The numbers are hard to tell what do each modules do exactly.

### questions
1. Just to make sure I understand correctly, the instance feature (100x249) and spatial feature (100x7) are fed into a MLP for fusion? Can you describe the archi?
2. Local spatial feature contains the normalized bounding box, confidence and top-rated semantic label. Is the semantic label the class index (1,2,...,C)? why not use a one-hot embedding or something?
3. Is AI2-THOR the most popular benchmark for object-goal nav? I have seen lots of prior paper running on habitat. What's the specific reasons of using AI2-Thor over habitat?

Please address my questions. I'm looking forward to discussing with the authors and the peer reviewers.

---

> ### Author Response · Authors · 2020-11-24
> **Further explanations on the feature extractors (i.e. DETR and Faster R-CNN) and experiments (1)**
>
> We thank you for acknowledging that our work is novel and experiments are sufficient. We also thank you for your constructive and detailed suggestions.
>
>
> **Q1: Details clarification**
>
> Thank you for your comments. We will add detailed notations for all features/descriptors in the figure captions and include the notations/details in Figure 2. We employ 1x1 convolution to reduce the channel dimension of instance features from 256 to 249. We will clarify the confusing statements in the revised version.
>
>
> **Q2: Extracted features from Faster R-CNN**
>
> Our Faster R-CNN instance features are obtained by following ORG (Du et al., 2020). The features are extracted from the second layer in the backbone based on the predicted bounding-boxes to ensure the features are compatible. However, the extracted features by ORG are not the most prominent ones across the feature pyramid. Thus, we argue that the object features are not scale-invariant but scale-sensitive. We will clarify this in the revision.
>
> Although on COCO validation set DETR and Faster R-CNN achieve similar detection performance (43.3% vs. 42.0% mAP), DETR reasons about the relations between object instances and the global image context via its transformer to output the final predictions (i.e. class labels and bounding boxes), as explained by Carion et al. (2020), while Faster R-CNN features do not involve image global context information. This motivates us to employ DETR features for navigation. Since features extracted by DETR are more informative and robust than those of Faster R-CNN used in ORG, our navigator employing DETR features outperforms ORG.
>
> Moreover, we also adopt the feature extraction manner used in Up-Down (Anderson et al., 2018) for navigation as another baseline, where object features are obtained from the penultimate layer of the classifier in Faster R-CNN. Therefore, the scales are invariant compared to ORG. As indicated in Table. 2, this baseline achieves the success rate of 57.2% and is still inferior to our baseline with DETR features. This indicates that DETR features are more informative as they not only contain object local appearances but also embed the relations between local features and global contexts. We will include these discussions and explanations in the revised version.
>
>
> **Q3: Explanation on early stopping**
>
> Due to the large parameters and complex architectures of transformers, it is often difficult to train transformers from scratch (Liu et al., 2020). Without a good initialization for our VT, it is very difficult to learn our VT and policy network in an end-to-end fashion with RL rewards. This is because the visual representations from randomly initialized VT are not informative or even meaningless and the inferior visual representations also handicap policy network learning. As a result, the navigation policy network may be trapped into a local minimum (i.e., terminating navigation early to avoid further penalties) and our VT cannot receive positive rewards from preceding trajectories. As seen in Figure 7, our navigator w/o pre-training scheme stops around 5 steps after training tens of thousands of episodes.
>
> To overcome this issue, we introduce our pre-training scheme. After warming-up, our VT embeds directional signals into visual representations and thus achieves a good initialization. Then, the positive rewards can lead our navigation system to learning an effective navigation policy.
>
> **Q4: Correction of SPL formulation**
>
> Thanks for pointing this out. We missed the success indicator in the SPL formulation, and will fix this in the revised version.
>
>
> **Q5: Adding global features into transformer encoder**
>
> Thank you for the suggestion. As suggested, We feed global features into the transformer encoder and then train the navigation network following our training protocols, marked as VTNet$_g$. The success rate ($\pm$ variance) of VTNet$_g$ is 70.1% ($\pm$ 1.3) while our VTNet achieves the success rate of 72.2% ($\pm$ 1.0). Meanwhile, the SPL of VTNet$_g$ is 0.411 ($\pm$ 0.009), and SPL of our VTNet is 0.449 ($\pm$ 0.007).
> By feeding the global features into the transformer encoder, the navigation performance of VTNet$_g$ is superior to that of ORG but slightly inferior to the performance of our VTNet. This demonstrates the transformer architecture is effective to extract informative visual representations, and assigning different functions to different modules would further facilitate the establishment of mappings in our VT. To be specific, our VT encoder focuses on establishing the correlations among detected instance features while our VT decoder associates object instance information to image regions to learn visual representations involving directional navigation signals.

---

> > ### Author Response · Authors · 2020-11-24
> > **Further explanations on the feature extractors (i.e. DETR and Faster R-CNN) and experiments (2)**
> >
> > **Q6: Experimental results with variance**
> >
> > Thanks for your suggestion. As demonstrated in Table 1, we report navigation results with the variances by repeating experiments five times. The success rate ($\pm$ variance) of our VTNet is 72.2($\pm$ 1.0), and Baseline is 62.6 ($\pm$ 0.9). This illustrates that our method is robust in unseen environments. We have updated the results with variances in the revision.
> >
> > **Q7: Additional questions on the MLP architecture, embedding, and testing environments**
> >
> > (a) After obtaining instance features (100x249), we develop spatial features (100x7). Each spatial feature includes a target indicator, a normalized bounding-box position, a confidence score and a predicted label from DETR. Then, we concatenate instance features (100x249) and spatial features (100x7) into the combined features (100x256). We employ two fully-connected layers with ReLU to fuse the combined features and then obtain our spatial-enhanced descriptors (100x256).
> > (b) From our experiments, we did not notice obvious performance differences between using one-hot embedding and using the class index.  However, as the number of categories increases, adopting one-hot embedding will increase the dimension of instance descriptors. Then, when we fuse the spatial features with instance features, we need MLPs (fully-connected layers) with larger parameters. This would lead to overfitting or require too much memory for training. Therefore, we select the class index as the semantic label.
> > (c) Since the state-of-the-art methods  (Wortsman et al., 2019; Du et al., 2020) conducted their experiments on AI2-Thor and released their training codes publicly, we select the AI2-Thor environment for fair comparisons. Moreover, AI2-Thor contains more objects of small sizes, such as cellphones and alarmcocks, thus increasing the difficulty of object goal navigation. Therefore, this environment is more suitable for testing the visual perception capability in object goal navigation tasks.
> >
> >
> > **Q8: Visualization of attention mechanism**
> >
> > Thank you for your suggestions. We provide more visualization results as seen in Figure 8. By visualizing the attention scores, we demonstrate that our VT indeed establishes relationships between object instances and directional signals that are highly correlated to image regions.
> >
> > **References**
> >
> > [1] Wortsman, M., Ehsani, K., Rastegari, M., Farhadi, A., & Mottaghi, R. (2019). Learning to learn how to learn: Self-adaptive visual navigation using meta-learning. In Proceedings of the IEEE Conference on Computer Vision and Pattern Recognition (pp. 6750-6759).
> >
> > [2] Du, H., Yu, X., & Zheng, L. (2020). Learning Object Relation Graph and Tentative Policy for Visual Navigation. arXiv preprint arXiv:2007.11018.
> >
> > [3] Carion, N., Massa, F., Synnaeve, G., Usunier, N., Kirillov, A., & Zagoruyko, S. (2020). End-to-End Object Detection with Transformers. arXiv preprint arXiv:2005.12872.
> >
> > [4] Anderson, P., He, X., Buehler, C., Teney, D., Johnson, M., Gould, S., & Zhang, L. (2018). Bottom-up and top-down attention for image captioning and visual question answering. In Proceedings of the IEEE conference on computer vision and pattern recognition (pp. 6077-6086).
> >
> > [5] Liu, L., Liu, X., Gao, J., Chen, W., & Han, J. (2020). Understanding the Difficulty of Training Transformers. arXiv preprint arXiv:2004.08249.

---

### Official Review · AnonReviewer1 · 2020-10-28
**An empirical study of usefulness of pre-training for navigation with transformers**

**Rating:** 6
**Confidence:** 4

**Review:**

This paper demonstrates a model that uses the Transformer to encode the visual features that appeared in the visual input image during navigation. The model is firstly pre-trained under imitation learning objective with self-generated shortest-path trajectories. The empirical results show that the model used in the paper outperforms previous methods on AI2-THOR environment. The authors also show some studies on the contributions of each component in the model.

Paper strengths:

+ The proposed method further show that the Transformer is a powerful model for feature extraction

+ The authors demonstrate one method to make the training of Transformer work, i.e. pre-training transformers using shortest-path trajectories

+ Empirical result support the authors' claims.

+ A thorough ablation study and discussions are provided.

Cons:

- The paper adopts the Transformer and adapted it into the navigation problem. No new architecture/model is proposed.

- It seems that a similar usage of Transformer already appeared in the vision-and-language navigation task [1]. The paper also shows that pre-training of navigation tasks using Transformers can help to boost the performance.

Minor: Two missing citations [2,3] that are potentially relevant.

[1] Towards Learning a Generic Agent for Vision-and-Language Navigation via Pre-training

[2] Evolving Graphical Planner: Contextual Global Planning for Vision-and-Language Navigation

[3] Are You Looking? Grounding to Multiple Modalities in Vision-and-Language Navigation

--

I've read the authors' response and would like to maintain my original score

---

> ### Author Response · Authors · 2020-11-24
> **Differences between our VTNet and Transformers used in VLN**
>
> We appreciate that R1 acknowledges the superiority of our performance and thorough experiments. We also thank you for your constructive and detailed comments.
>
>
> **Q1: Our proposed Visual Transformer (VT)**
>
> To the best of our knowledge, our method is the first attempt to adopt the transformer for object goal navigation. Furthermore, due to the differences between object goal visual navigation and vision-and-language navigation (VLN), our proposed Visual Transformer (VT) has different designs from the transformers used in VLN.
> We proposed a visual transformer (VT) to explore the spatial and category relations among all detected instances as well as leverage correspondences between instances and image regions to generate informative visual representations for navigation. However, in the VLN task, the transformers in Prevalenet (Hao et al., 2020) are used to model languages by masking words and select actions by predicting a preferred field of view (FoV) from a panorama view. Thus, our VT is essentially different from the transformers used in VLN.
>
> **Q2: Differences from Pre-training Transformers used in VLN**
>
> We agree with R1 that pre-training is used to warm-up transformers for navigation. However, we would like to point out that due to the different designs and tasks between object goal navigation and VLN, the pre-training schemes impact differently on the transformers of VLN and our VT. To be specific, the VLN transformers in Prevalenet are pre-trained by masking out words in navigation instructions and predicting camera angles. As reported by Prevalenet, the performance improvements mainly come from the language transformer pre-training, which adapts the transformer to the navigation instructions, while the action prediction pre-training only improves the final performance slightly. This might be because the language instructions dominate the VLN process. In contrast to VLN, the visual representations are more critical in object goal visual navigation. Thus, our VT pre-training scheme establishes associations between visual representations and directional signals via shortest-path demonstrations. As indicated by the experimental results, our pre-training scheme not only boosts our performance but also is essential in training our VT.
>
> **Q3: Missing references**
>
> Thanks for pointing out the missing references. We have cited them in our revised version.
>
> **References**
>
> [1] Hao, W., Li, C., Li, X., Carin, L., & Gao, J. (2020). Towards learning a generic agent for vision-and-language navigation via pre-training. In Proceedings of the IEEE/CVF Conference on Computer Vision and Pattern Recognition (pp. 13137-13146).

---

### Official Review · AnonReviewer3 · 2020-10-29
**Novel approach for learning visual representations for navigation; but weak experiments and explanations.**

**Rating:** 6
**Confidence:** 4

**Review:**

Summary
The paper proposes Visual Transformer Network which encodes the relationship between all detected object instances in a frame and uses it for navigation. The paper uses DETR for object detection and learn an association between local descriptors (from the object detector) with global descriptors (ResNet18) using the proposed VT model. They show that using VT improves performance on the object navigation task in AI2-THOR simulator compared to existing methods.

Strengths
- The paper proposed a novel transformer architecture that learns an association between local object descriptors with global image region features so that actions can be grounded to visual regions in the image.

- Different from prior work, the paper uses all the objects detected for a label instead of just the most confident detection.


Weaknesses

- The paper doesn't fully address why DETR performs better than FasterRCNN features. Appearance features from FasterRCNN have been widely used for several downstream tasks in Vision and Language Navigation[1], Vision and Language tasks[2]. From the experiments, it's not clear why DETR is doing better than Faster-RCNN especially when the detection accuracy of DETR is also better than Faster RCNN.

- Additionally, I didn't fully follow how authors obtain the appearance features from Faster RCNN based method. The authors mention that object appearance features are extracted from different layers of a backbone network. How is it different from the approach taken by Bottom-Up, Top-Down[3] paper in which 2048-dim appearance features are extracted for each visual region?

- The experimental setup isn't fully reflective of the object goal navigation task. The experiments are conducted in AI2 thor scenes which only contain one room. It's not clear, how this method will perform when evaluated on significantly more complicated environments like Matterport / Gibson [4]. Specifically, I am interested in how will the proposed architecture perform when the goal object is not in the same room as the agent.

- The navigation task is also made simpler by discretizing into a grid. Single room environments and discrete grids simplify a lot of navigation-related challenges and the authors don't discuss how the proposed architecture will generalize to more complex object navigation tasks.

- The use of spatial embeddings as well as appearance embedding isn't all that surprising. Existing work including Du et al. uses bounding box coordinates to help learn spatial associations between objects.

Other questions:
- Instead of pre-training without employing the navigation policy, did the authors try using shortest-path based demonstrations to help learn the navigation policy as well? In the first stage, the navigation policy learns using imitation learning and then finetuned with A3C?

- What is the step size of the agent for the forward step? What are the turn angles for Turn-left, Turn-right actions? What are the tilt angles for look-up and look-down actions?

- What's the reason for improvement over ORG (in absence of TPN). Is it superior visual representations (Faster RCNN vs DETR) or the fact ORG only chooses objects with the highest confidence while VT uses all the detected objects?

- How does the agent learn long-term associations between objects across multiple frames. In my opinion, the proposed architecture puts all the burden of learning these long-term object relationships across multiple frames on the LSTM policy since the VT only learns association within a single frame.

[1] Improving Vision-and-Language Navigation with Image-Text Pairs from the Web; Majumdar et al.

[2] Oscar: Object-Semantics Aligned Pre-training for Vision-Language Tasks; Li et al.

[3] Bottom-Up and Top-Down Attention for Image Captioning and Visual Question Answering; Anderson et al.

---

> ### Author Response · Authors · 2020-11-24
> **More explanations on the feature extractors (i.e., DETR and Faster R-CNN) and experiment setting (1)**
>
> We thank you for acknowledging the novelty of our work. We also thank you for the constructive and detailed comments.
>
> **Q1: Features from DETR outperform those of Faster R-CNN**
>
> Unlike Faster R-CNN, DETR (Carion et al., 2020) infers the relations between object instances and the global image context via its transformer to output the final predictions (i.e. class labels and bounding boxes), as explained by Carion et al. (2020). This motivates us to employ DETR features for navigation.
>
> Although on COCO validation set DETR and Faster R-CNN achieve similar detection performance (43.3% vs. 42.0% AP), features extracted by DETR are more informative and robust than those of Faster R-CNN used in ORG (Du et al., 2020). Note that Faster R-CNN features are obtained by following ORG. Specifically, the features are extracted from the second layer in the backbone based on the predicted bounding-boxes to ensure the features are comparable, but the features are not the most prominent ones across the feature pyramid. Therefore, the object features extracted by Faster R-CNN are inferior to DETR features, and our navigator employing DETR features outperforms ORG.
>
> We thank R3 for pointing out the feature extraction differences between ORG and Up-Down (Anderson et al., 2018). It would be another option to attain features of Faster R-CNN from the penultimate layer of the classifier in Faster R-CNN as in Up-Down. We then use these Faster R-CNN features for navigation, marked as Faster R-CNN$^\dagger$ in Table 2, and achieve the success rate of 57.2% in our baseline model, whereas using DETR features the success rate is 62.6%. Still, Faster R-CNN features do not take global context information into account and thus are not as informative as our DETR features. We have added this new experiment in our revision.
>
> Moreover, the recent papers (Li et al., 2020; Majumdar et al., 2020) do not use DETR for navigation because these papers and DETR are all published in ECCV 2020.
>
>
> **Q2: Details of ORG feature extraction**
>
> In this paper, we follow the feature extraction fashion provided by ORG. To be specific, the local features of object instances are extracted from the second layer in the backbone based on the predicted bounding-boxes to ensure the features are compatible. However, we argue that the object features extracted by ORG are not the most prominent ones across the feature scales.
>
> On the other hand, Up-Down (Anderson et al., 2018) obtains object features from the penultimate layer of the classifier in Faster R-CNN. Therefore, the features are scale-invariant compared to ORG. We also adopt the feature extraction fashion of Up-Down for navigation, marked as Faster R-CNN$^\dagger$ in Table 2. Replacing DETR features with Faster R-CNN$^\dagger$ features leads to 5.4% performance degradation in the baseline models. This indicates that DETR features are more informative as they not only contain object local appearances but also embed the relations between local features and global contexts.
>
>
> **Q3: Experiment setups**
>
> We follow the state-of-the-art visual navigation methods (Wortsman et al., 2019; Du et al., 2020) to conduct our experiments on the AI2-Thor simulator. In this simulator, there are some environments containing multiple rooms. For instance, as illustrated in Figure 3, two rooms are separated by a wall and an agent needs to pass through the door to reach the target in another room. In this case, our VTNet successfully finds the target object with the shortest steps. This indicates that our agent is able to navigate towards target objects across rooms when the targets are not in the same room as the agent.
>
> The targets in Habitats (Savva et al., 2019) are usually large and obvious objects, such as cabinets and tables. Compared to Habitat, AI2-Thor contains more object categories and target objects are usually in small sizes, such as cellphones and alarmcocks. In fact, AI2-Thor increases the difficulty of object goal navigation. Therefore, this environment is more suitable for testing the visual perception capability in object goal navigation tasks.
>
> Following the prior works (Wortsman et al., 2019; Du et al., 2020), we discretize each scene into grid points. Given the discrete grid graph, the next states of an agent are determined. Therefore, navigation in this environment can reflect the intention of an agent. In other words, the movements of an agent are fully determined by the chosen actions. On the contrary, in continuous environments, an agent can overcome obstacles by decomposing intentional movements into movements in occluded directions and ``unintentional’’ movements in unoccluded directions. The unintentional movements would help an agent to overcome the obstacles, thus increasing its success rates, However, unintentional movements will not happen in discretized environments. Therefore, navigation in discretized environments should not be regarded as an easy scenario.

---

> > ### Author Response · Authors · 2020-11-24
> > **More explanations on the feature extractors (i.e., DETR and Faster R-CNN) and experiment setting (2)**
> >
> > **Q4: Comparison with ORG (Du et al., 2020)**
> >
> > ORG models spatial and category closeness via MLPs, and selects the top-confidence candidates in each category. In this way, ORG may neglect some other instances for navigation and also be affected if top-confidence candidates are false positive. In contrast, our VT explores the relationship among all the detected instances. Meanwhile, we associate image regions and object instances to establish strong correlations between visual representations and navigation directional signals. As seen in Table 2, our DETR features significantly alleviates the shortcomings in ORG and thus our VTNet outperforms ORG.
> >
> >
> > **Q5: Our pre-training scheme**
> >
> > In our VT pre-training stage, we adopt the shortest-path based demonstrations and replace the navigation policy network with MLPs. In other words, the navigation policy network is not used in pre-training. In the fine-tuning stage, we remove MLPs and adopt our navigation policy as our navigator. Then, we fine-tune our VT with the reinforcement learning rewards.
> >
> >
> > **Q6: Implementation details**
> >
> > Following Wortsman et al. (2019) and Du et al. (2020), the forward step size is 0.25 metres, and the angles of turning-left/right and looking-up/down are 45$^\circ$ and 30$^\circ$, respectively. We will add these details in the revised version.
> >
> > **Q7: Improvement over ORG**
> >
> > Thank you for the question. We believe that our improvements over ORG come from two aspects. (1) Fully exploiting detection information: ORG only uses detected objects with the highest confidence while VT uses all the detected objects. Thus, our VT not only fully exploits information from all the instances but also alleviates the impacts of selecting false positive results compared to ORG. Similar to ORG, we use the detected candidates with top-confidences from DETR as the input for our VT, and the success rate decreases 1.2%, as seen in Table 2. This indicates the importance of exploiting all the instance information for navigation.
> > (2) Superior visual feature encoding: local object features extracted by DETR not only contain local object information but also global image context due to its transformer architecture. Moreover, as ORG extracts object features from the second layer in the backbone, the extracted object features may not be prominent across the feature pyramid. On the contrary, the features output by DETR are scale-invariant. Therefore, we improve the visual representations from both aspects and achieve better navigation performance.
> >
> >
> > **Q8: Long-term associations**
> >
> > Similar to state-of-the-art methods (Wortsman et al., 2019; Du et al., 2020), our navigation policy network is based on LSTMs and during navigation the policy network mainly focuses on predicting actions for next steps rather than establishing object associations across frames. Thank you for the constructive suggestions. We will investigate how to employ previous frames to improve navigation performance in our future work.
> >
> > **References**
> >
> > [1] Carion, N., Massa, F., Synnaeve, G., Usunier, N., Kirillov, A., & Zagoruyko, S. (2020). End-to-End Object Detection with Transformers. arXiv preprint arXiv:2005.12872.
> >
> > [2] Du, H., Yu, X., & Zheng, L. (2020). Learning Object Relation Graph and Tentative Policy for Visual Navigation. arXiv preprint arXiv:2007.11018.
> >
> > [3] Wortsman, M., Ehsani, K., Rastegari, M., Farhadi, A., & Mottaghi, R. (2019). Learning to learn how to learn: Self-adaptive visual navigation using meta-learning. In Proceedings of the IEEE Conference on Computer Vision and Pattern Recognition (pp. 6750-6759).
> >
> > [4] Anderson, P., He, X., Buehler, C., Teney, D., Johnson, M., Gould, S., & Zhang, L. (2018). Bottom-up and top-down attention for image captioning and visual question answering. In Proceedings of the IEEE conference on computer vision and pattern recognition (pp. 6077-6086).
> >
> > [5] Savva, M., Kadian, A., Maksymets, O., Zhao, Y., Wijmans, E., Jain, B., ... & Parikh, D. (2019). Habitat: A platform for embodied ai research. In Proceedings of the IEEE International Conference on Computer Vision (pp. 9339-9347).
> >
> > [6] Majumdar, A., Shrivastava, A., Lee, S., Anderson, P., Parikh, D., & Batra, D. (2020). Improving Vision-and-Language Navigation with Image-Text Pairs from the Web. arXiv preprint arXiv:2004.14973.
> >
> > [7] Li, X., Yin, X., Li, C., Zhang, P., Hu, X., Zhang, L., ... & Choi, Y. (2020, August). Oscar: Object-semantics aligned pre-training for vision-language tasks. In European Conference on Computer Vision (pp. 121-137). Springer, Cham.

---

### Author Response · Authors · 2020-11-25
**Revision and generic comment**

We thank the reviewers for their helpful comments and insightful suggestions. We carefully revised the manuscript according to the comments of all the reviewers. For convenience, we highlighted the revised text in color.

In this revision we have added:
- experiments using instance features extracted by Faster R-CNN following Up-Down (Anderson et al., 2018). We further provide some insights of the difference between DETR and Faster R-CNN through navigation performances.
- variances of navigation performance in Table 1 and Table 2.
- an experiment by adding global features into the encoder and more visualization results of the attention mechanism. We demonstrate that our VT is able to explore the relations between detected instances and image regions.
- discussion on the improvement of our pre-training scheme. We provide episode length curves in training and some insights on early stopping when we do not adopt the pre-training scheme.

Other revised parts:
- We provide the high-level overview of our method in the introduction and resolve the grammar and typo issues.
- We include the missing references and explain the differences between our VTNet and other transformers used in VLN in the Sec. 2.
- We provide detailed settings of agent actions in the Sec. 3.
- We clarify notations for all features/descriptors and captions of Figure 2 and Figure 4.
- We add more details about our proposed method and correct some typos in the Sec. 4.
- We clarify our positional embedding function by editing the Eq. 1.
- We correct typos in the SPL formulation.
- We provide further discussion in the failure case study.


**References**

[1] Anderson, P., He, X., Buehler, C., Teney, D., Johnson, M., Gould, S., & Zhang, L. (2018). Bottom-up and top-down attention for image captioning and visual question answering. In Proceedings of the IEEE conference on computer vision and pattern recognition (pp. 6077-6086).

---

### Decision · Program_Chairs · 2021-01-07
**Final Decision**

**Decision:**

Accept (Poster)

**Comment:**

This paper addresses the problem of visual object navigation by defining a novel visual transformer architecture, where an encoder consisting of a pretrained object detector extracts objects (i.e. their visual features, position, semantic label, confidence) that will serve as keys in an attention-based retrieval mechanism, and a decoder computes global visual features and positional descriptors as a coarse feature map. The visual transformer is first pretrained (using imitation learning) on simple tasks consisting in moving the state-less agent / camera towards the target object. Then an RL agent is defined by adding an LSTM to the VTNet and training it end-to-end on the single-room subset of the AI2-Thor environment where it achieves state-of-the-art performance.

After rebuttal, all four reviewers converged on a score of 6. The reviewers praised the novelty of the method, extensive evaluation with ablation studies, and the SOTA results. Main points of criticism were about clarity of writing and some explanations (which the authors improved), using DETR vs. Faster R-CNN, and the relative simplicity of the task (single room and discrete action space). There were also minor questions, a request for more recent transformer-based VLN bibliography, and a request for a new evaluation on RoboThor. One area of discussion -- where I empathise with the authors -- was regarding the difficulty of pure RL training of transformer-based agents and the necessity to pre-train the representations.

Taking all this into account, I suggest this paper gets accepted.